# Patients with genetically heterogeneous synchronous colorectal cancer carry rare damaging germline mutations in immune-related genes

Matteo Cereda[1,*], Gennaro Gambardella[1,*], Lorena Benedetti[1,*], Fabio Iannelli[2], Dominic Patel[3], Gianluca Basso[4], Rosalinda F. Guerra[5], Thanos P. Mourikis[1], Ignazio Puccio[3], Shruti Sinha[1], Luigi Laghi[4], Jo Spencer[6], Manuel Rodriguez-Justo[3] & Francesca D. Ciccarelli[1]

Synchronous colorectal cancers (syCRCs) are physically separated tumours that develop simultaneously. To understand how the genetic and environmental background influences the development of multiple tumours, here we conduct a comparative analysis of 20 syCRCs from 10 patients. We show that syCRCs have independent genetic origins, acquire dissimilar somatic alterations, and have different clone composition. This inter- and intratumour heterogeneity must be considered in the selection of therapy and in the monitoring of resistance. SyCRC patients show a higher occurrence of inherited damaging mutations in immune-related genes compared to patients with solitary colorectal cancer and to healthy individuals from the 1,000 Genomes Project. Moreover, they have a different composition of immune cell populations in tumour and normal mucosa, and transcriptional differences in immune-related biological processes. This suggests an environmental field effect that promotes multiple tumours likely in the background of inflammation.

[1] Division of Cancer Studies, King's College London, London SE1 1UL, UK. [2] IFOM, FIRC Institute of Molecular Oncology, Milan 20139, Italy. [3] Department of Research Pathology, Cancer Institute, University College London, London WC1E 6JJ, UK. [4] Laboratory of Molecular Gastroenterology, Department of Gastroenterology, Humanitas Research Hospital, Rozzano (MI) 20089, Italy. [5] Department of Craniofacial Development & Stem Cell Biology, King's College London, London SE1 9RT, UK. [6] Department of Immunobiology, King's College London, London, SE1 9RT, UK. * These authors contributed equally to this work. Correspondence and requests for materials should be addressed to F.D.C. (email: francesca.ciccarelli@kcl.ac.uk).

Several large-scale sequencing projects have extensively characterized the genomic landscape of colorectal cancer (CRC)[1–4]. Despite all efforts, several questions still remain unaddressed. For instance, around 2–5% of CRC patients present multiple primary tumours at initial diagnosis[5–7] (synchronous CRC (syCRC)) but the causes of multiple tumours are still poorly understood. Patients with Lynch syndrome and familial adenomatous polyposis (FAP) have a higher incidence of syCRC[8–10]. Similarly, inflammatory bowel diseases (IBDs) and hyperplastic polyposis are known to predispose to synchronous tumours[10–12]. These conditions, however, only account for around 10% of syCRC[8], thus suggesting that other predisposing causes likely exist[7,13]. Recently, a homozygous mutation in the base-excision repair gene *NTHL1* has been associated with the onset of multiple colorectal adenomas in Dutch families[14], but this mutation is absent in other affected individuals. In addition to the predisposing factors, it is uncertain whether paired tumours of a patient share the same genetic origin and acquire similar somatic alterations. In other words, whether genetic or environmental field effects influence the way syCRCs originate and develop. Comparative analyses of syCRCs have so far focussed mostly on mutation hotspots in known cancer genes and on the status of microsatellites and mismatch-repair proteins. These studies report both concordant and discordant alterations between paired tumours, with the latter being predominant[15–21]. High methylation of CpG islands seems to be a recurrent feature of syCRC[22–24] and has suggested the presence of an epigenetic field effect[13,23]. Despite these reports, a comprehensive characterization of syCRC is not yet available. Around 5% of CRCs available in the Cancer Genome Atlas (TCGA) are annotated as synchronous, but only one tumour has been sequenced in all cases, thus preventing a genome-wide comparison of somatic mutations between paired lesions.

Here, we performed a systematic genomic profiling of 20 syCRCs from 10 patients with the aim to compare their alterations. The analysed cohort included patients with Lynch syndrome, FAP, Peutz–Jeghers syndrome, familial CRC type X and sporadic CRC, not to bias the analysis towards a particular CRC type. We compared the landscape of acquired mutations between paired tumours to assess whether they share the same origin and converge towards a similar clone composition. We also analysed the inherited genotype of these patients to search for evidence of genetic predispositions to the development of multiple tumours. Our results contribute the elucidation of the genetics and of the predisposing mechanisms of syCRCs with possible impacts on their clinical management.

## Results

**syCRCs are genetically heterogeneous tumours.** We extracted genomic DNA from multiple sections of one fresh frozen and 19 formalin-fixed paraffin embedded (FFPE) tumours from 10 patients (Supplementary Table 1). Each tumour underwent quantitative pathological review to ensure a reliable estimation of tumour content based on macrodissected sections or across multiple regions of the tissue block (Supplementary Fig. 1). We captured and sequenced the whole exomes of all 20 syCRCs and matched normal samples reaching an average depth of coverage of $125 \times$ (Supplementary Table 2; Supplementary Fig. 2). We called and compared single nucleotide variants (SNVs) and insertions and deletions (InDels) in tumour and corresponding normal (Supplementary Fig. 3) to identify somatic mutations (Table 1; Supplementary Data 1). We performed several quality controls on the identified mutations. First, we re-sequenced the whole exome of eight tumours from independent libraries (Supplementary Fig. 2) and confirmed that on average

81% mutations were present in both sequencing rounds (Supplementary Table 3). Second, we re-called SNVs and InDels using independent variant callers, and measured 88 and 78% concordance, respectively (Supplementary Table 3). Third, we re-sequenced a panel of 151 cancer genes in six tumours at high depth of coverage ($280 \times$, Supplementary Table 4) and confirmed all previously detected mutations (Supplementary Data 2). Finally, we randomly selected 24 SNVs and 11 InDels, and confirmed 32 of them with Sanger sequencing (91% specificity, Supplementary Table 5). Since the majority of sequenced samples derived from FFPE tissues, we checked for possible sequencing artefacts due to formalin fixation. We observed similar signatures of somatic mutations between the 19 FFPE tumours and fresh frozen CRCs from TCGA (Supplementary Fig. 4). Similarly, the germline mutation patterns of FFPE samples were comparable to those of blood and TCGA samples (Supplementary Fig. 4). This excluded the presence of sequence artefacts.

In addition to SNVs and InDels, we also profiled copy number variations (CNVs) using genome-wide SNP array on tumours and normal samples, and identified genes undergoing somatic amplifications and deletions in each tumour (Table 1; Supplementary Data 3). As expected[1,25], cancers associated with mismatch repair deficiency were hypermutated and mostly diploid, while non-hypermutated CRCs showed a high proportion of amplified or deleted genes (Fig. 1a; Table 1). We also confirmed that cancers on the right side of the intestine were more mutated than those on the left side[1,25] (Fig. 1a).

To assess whether syCRCs shared the same genetic origin, we compared somatic alterations between paired tumours. Overall, we detected high intertumour heterogeneity, with almost all nonsilent mutations being dissimilar between lesions (Fig. 1b). To discard the possibility that mutations were not identified because of insufficient coverage depth, we verified that all mutated positions in one tumour were well covered and wild type in the other (Supplementary Fig. 5). Similar to mutations, most genes underwent different types of CNVs in paired tumours (Fig. 1c). The exceptions were patients S3, S12 and UH5 whose tumours shared the same amplified regions on chromosomes 1 and 3. However, these CNVs had different breakpoints (Supplementary Fig. 6), suggesting that they occurred independently in each tumour.

Next we investigated whether, although with different modifications, tumours of a patient converged towards the modification of the same cancer genes. Overall, only a small fraction of cancer genes were altered in both tumours (Fig. 1d; Supplementary Data 4). This was further confirmed in the deep sequencing experiment (Supplementary Data 2). To assess whether the few shared cancer genes were altered at similar stages during the tumour growth, we quantified the clonality of their alterations (see below). In the majority of cases, alterations in cancer genes had different clonality (Fig. 2a), indicating that the corresponding cancer genes were modified at different times in the two tumours. For example, alterations in *PIK3CA* and *ARID1A* were early events in tumour S13T2 (clonality >60%), but were detectable in <35% of cells in S13T1 (Fig. 2a). Patients S3, S12, and to a lower extent, UH5 again represented exceptions because their tumours shared higher fractions of altered cancer genes (Fig. 1d) and these alterations, albeit different (Supplementary Fig. 6), often had similar clonality (Fig. 2a). Finally, we checked whether, overall, tumours of a patient converged towards the alteration of the same genes. We found that paired lesions of a patient did not share a higher fraction of altered genes or of cancer genes than any pair of tumours of different patients (*P* values = 0.34 and 0.16, respectively, one-tailed Wilcoxon rank-sum test, Fig. 2b,c). Altogether our analysis indicated that syCRCs had independent genetic origins, acquired

**Table 1 | Somatic nonsilent mutations and copy-number variant genes in syCRCs.**

| Patient ID | Gender | Age at diagnosis (y/o) | Cancer type | Germline mutation | Tumour | Nonsilent mutations (n) | CNV genes (n) |
|---|---|---|---|---|---|---|---|
| S13 | F | 37 | Lynch syndrome | MLH1: p.R100* | T1 | 523 | 6,059 |
| | | | | | T2 | 654 | 1,094 |
| S6 | M | 40 | Lynch syndrome | MSH2: p.Q718* | T1 | 1,150 | 3,546 |
| | | | | | T2 | 478 | 2 |
| S3 | F | 29 | FAP | APC: p. I1307K | T1 | 47 | 12,458 |
| | | | | | T2 | 52 | 9,222 |
| S12 | M | 80 | Sporadic | — | T1 | 72 | 14,109 |
| | | | | | T2 | 61 | 14,448 |
| UH1 | M | 66 | PJS | STK1: p.F354L | T1 | 1,232 | 7,448 |
| | | | | | T2 | 34 | 12,647 |
| UH2 | F | 66 | Sporadic | — | T1 | 822 | 2,429 |
| | | | | | T2 | 281 | 2 |
| UH5 | F | 70 | Sporadic | — | T1 | 569 | 8,788 |
| | | | | | T2 | 89 | 11,461 |
| UH6 | M | 86 | FCCTX | SEMA4A: p.P682S | T1 | 26 | 126 |
| | | | | | T2 | 75 | 1,013 |
| UH8 | M | 69 | Sporadic | — | T1 | 59 | 5,412 |
| | | | | | T2 | 37 | 11,029 |
| UH11 | M | 65 | Lynch syndrome | MLH1: p.G67R | T1 | 515 | 0 |
| | | | | | T2 | 1,021 | 658 |

FAP, familial adenomatous polyposis; FCCTX, familial CRC type X; PJS, Peutz–Jeghers syndrome.
Reported for each patient are the number of somatic nonsilent mutations (SNVs and InDels) and copy number variant (CNV) genes. Germline predisposing mutations are described according to the Human Genome Variation Society (http://www.hgvs.org/mutnomen).

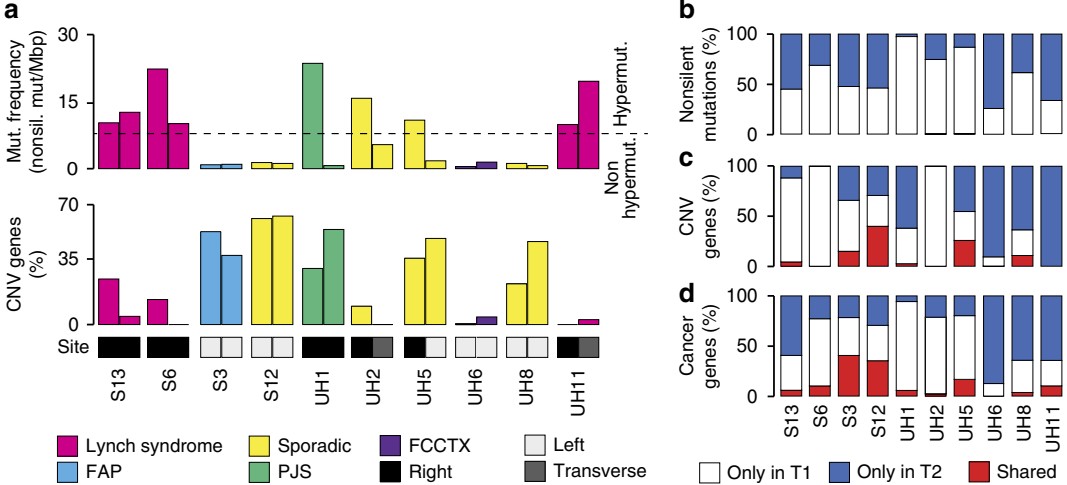

**Figure 1 | Landscape of somatic alterations in syCRC.** (**a**) Mutation frequency and percentage of amplified and deleted genes in 20 syCRCs from 10 patients. The dashed line represents the threshold of mutation frequency (eight mutations per mega base pairs) between hypermutated and non-hypermutated CRC[1]. FAP, familial adenomatous polyposis; FCCTX, familial CRC type X; PJS, Peutz–Jeghers syndrome. (**b–d**) Percentage of somatic nonsilent mutations, copy number variant genes, and altered cancer genes that are shared between paired tumours or private to one of them.

different somatic alterations and developed into tumours that were as genetically heterogeneous in terms of cancer genes as tumours from different patients.

**syCRCs show distinct clone composition.** Next, we sought to investigate whether syCRCs were also heterogeneous in terms of clone composition, because this may have consequences in their clinical management. To this aim, we derived the density distribution of clonality of somatic alterations and quantified the number of prevalent clones in each tumour using the allele frequency of somatic SNVs, InDels, amplifications and deletions (Fig. 3a). For SNVs and InDels, we measured the allele frequency as the number of mutated reads over the total reads, after confirming the reliability of this estimation (Supplementary Fig. 7). Because amplifications and deletions modify the allele frequency

of SNVs and InDels, we only considered mutations in diploid regions (on average 76% of all somatic mutations in each tumour, Supplementary Data 1). For amplified and deleted regions, we inferred the allele frequency based on the loss of heterozygosity of germline mutations (see Methods section). We corrected the allele frequency of each alteration for the tumour content of the corresponding lesion to remove the fraction of wild-type alleles deriving from normal cells (Fig. 3a). We then assessed the fraction of cancer cells carrying each alteration (alteration clonality) and derived the density distributions of clonality for SNVs, InDels, amplifications and deletions independently (Fig. 3a). In general, these density distributions indicated how many alterations were expected at each clonality as inferred from the observed counts. In particular, the peaks of the distributions showed at which clonality the alterations accumulated and were indicative of the clone composition of the tumour. To further

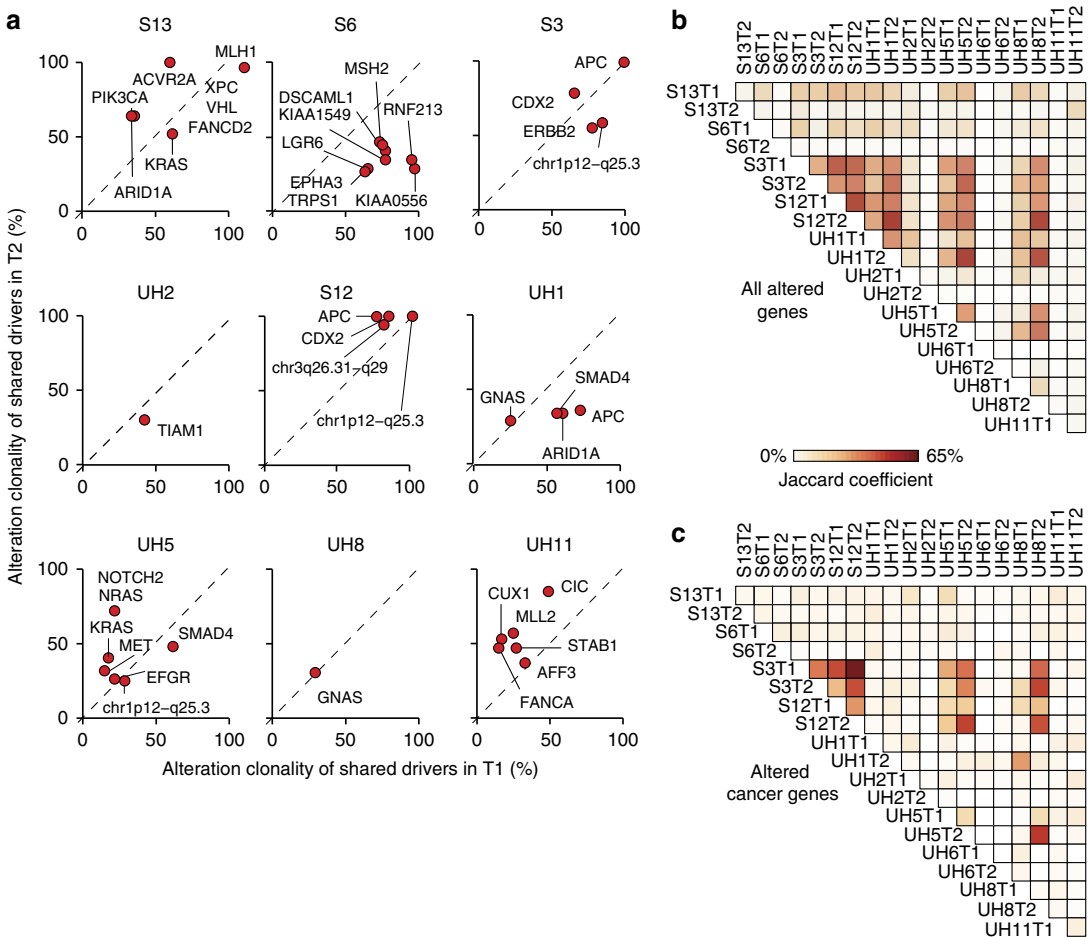

**Figure 2 | Comparison of somatic alterations between paired tumours. (a)** Clonality of putative driver alterations in cancer genes that are shared between paired tumours. Alteration clonality corresponds to the fraction of cancer cells carrying the alteration (see Methods section). Tumours of patient UH6 are not shown because they did not share any altered cancer genes (Supplementary Data 4). **(b,c)** Similarity matrix of all altered genes **(b)** and of altered cancer genes **(c)** across the 20 syCRCs. For each pair of tumours from same or different patients, the Jaccard coefficient was measured as the proportion of shared altered genes over the total number of altered genes.

classify the tumour as monoclonal, biclonal or polyclonal, we divided somatic alterations into three groups according to their clonality ($>80\%$, 35–80% and $<35\%$). The largest group of the three was indicative of the presence of one, two or multiple prevalent clones (Fig. 3a; Supplementary Note 1). Three out of 20 tumours (S3T1, S12T1 and S12T2) were classified as monoclonal because the majority of their somatic events were detectable in $>80\%$ of cancer cells (Fig. 3b; Supplementary Fig. 8). Ten tumours (S13T1, S13T2, S6T1, S3T2, UH1T1, UH2T1, UH5T1, UH5T2, UH8T1 and UH11T2) were considered as biclonal because they showed an accumulation of modifications between 35 and 80% clonality, suggesting the co-existence of two prevalent clones (Fig. 3b; Supplementary Fig. 8). The remaining seven tumours (S6T2, UH1T2, UH2T2, UH6T1, UH6T2, UH8T2 and UH11T1) were considered as polyclonal because the majority of somatic events had clonality $<35\%$, compatibly with the presence of multiple clones (Fig. 3b; Supplementary Fig. 8). In 6 out of 10 patients, the two tumours had distinct clone composition (Fig. 3b), indicating that syCRCs differed not only in their genetic origin but also in their clonal development. Remarkably, we obtained similar estimates of clone composition using an independent method to measure the alteration clonality[26] (Supplementary Fig. 9).

The inter- and intratumour heterogeneity of syCRCs may have implications in the response of patients to therapy. For example,

several analysed tumours had amplification of EGF receptor (EGFR) (Fig. 3c; Supplementary Data 4), which is a clinically relevant target in CRC[27–29]. However, they also showed activating alterations of EGFR downstream effectors conferring resistance to anti-EGFR therapy[30–33] (Fig. 3c; Supplementary Data 4). In most cases, these alterations were heterogeneously distributed at different clonality between paired tumours (Fig. 3c; Supplementary Fig. 8). For example, modifications of *EGFR* and *PIK3CA* were almost clonal in tumour S6T1 but absent in S6T2 (Fig. 3d). We extended our analysis to a list of known actionable genes[34] and again observed that most alterations either occurred in only one tumour or showed different clonality between tumours (Supplementary Fig. 10). Therefore, the different clone composition of syCRCs might have an impact on drug response and their genetic heterogeneity should be taken into account when selecting therapeutic regimens.

**syCRC patients carry damaging germline SNPs in immune genes.** Our analysis of somatic alterations showed that syCRCs started independently and developed into heterogeneous tumours as a consequence of distinct driver events. Thus, we asked whether there was any genetic predisposition of these individuals to develop independent tumours as compared to patients with solitary CRC (soCRC). We identified 406 soCRCs and 23

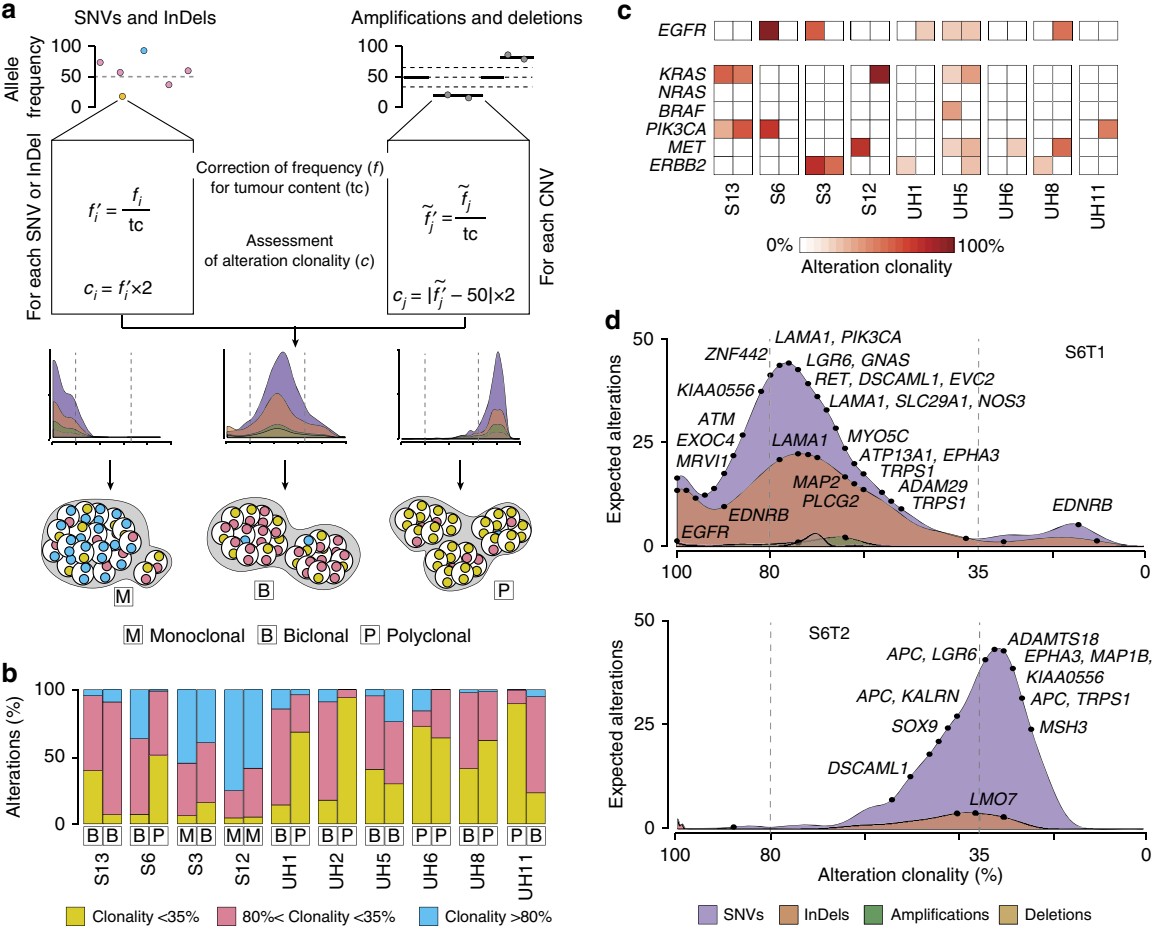

**Figure 3 | Inter and intratumour heterogeneity of syCRCs.** (**a**) Method to rebuild the clone composition of a tumour. Allele frequency of each somatic alteration is corrected for the tumour content and used to infer the alteration clonality. The density distributions of clonality for each type of alteration recapitulate the tumour clone composition (monoclonal, if the majority of alterations has clonality >80%; biclonal, if alterations accumulate between 35 and 80% clonality; or polyclonal if they have <35% clonality). (**b**) Clone composition of the 20 syCRCs as inferred from the clonality of their somatic alterations. (**c**) Clonality of putative driver alterations in *EGFR* and in other six genes known or suspected to give resistance to anti-EGFR therapy. Data are shown in both lesions of all patients, except UH2 because neither tumour from this patient showed alterations in the EGFR pathway. (**d**) Density distributions of clonality for somatic alterations in the two tumours of patient S6. Dots represent alterations in CRC genes as collected from the Network of Cancer Genes[61].

additional syCRCs in TCGA and verified that overall the two cohorts did not significantly differ in terms of age at initial diagnosis, gender, ethnicity and CRC type (Supplementary Fig. 11). The only significant difference was the higher occurrence of extra-colonic malignancies in syCRC patients (*P* value = 0.03, Fisher's exact test, Supplementary Fig. 11), which supports the hypothesis of their predisposition to develop multiple tumours.

Since known hereditary conditions account only for a small fraction of syCRC[8], we hypothesized that syCRCs result from the constitutional alteration not of a single gene, but of several genes contributing to the same biological process. To detect such altered processes, we developed a mutation enrichment gene set analysis (MEGA). MEGA systematically compares the cumulative distribution of mutations within a process between two cohorts and identifies those processes that are overall more frequently altered in one cohort (Fig. 4a, see Methods section). Using MEGA, we compared the distribution of rare single nucleotide polymorphisms (SNPs) with predicted damaging effects on the protein in 186 manually annotated KEGG gene sets[35] between syCRC and soCRC patients. We focused on rare damaging SNPs (minor allele frequency <1%) because they are most likely to

cause disease[36]. Since the TCGA samples were sequenced at different centres, we re-called mutations using the same pipeline as for our samples (Supplementary Data 5). In syCRC patients, we observed a significantly higher number of rare damaging SNPs in four of the 186 KEGG gene sets (cytokine-cytokine receptor interaction, Toll-like receptor signalling, biosynthesis of unsaturated fatty acids and cytosolic DNA sensing pathways, false discovery rate, FDR <10%, Supplementary Data 6). To exclude possible biases, we repeated the analysis using different reference cohort and gene sets. To change cohort, we re-called and annotated germline mutations in 756 individuals of the 1,000 Genomes Project[37] (Supplementary Data 5). We again confirmed the enrichment of syCRC patients in rare damaging SNPs affecting the cytokine-cytokine receptor interaction and in the Toll-like receptor signalling pathways (FDR <2%, Fig. 4b; Supplementary Data 6). To change gene sets, we grouped 6,589 disease-associated genes[38] according to disease and obtained 346 disease-associated gene sets (Supplementary Data 7). Applying MEGA we found that syCRC patients had significantly more rare damaging SNPs in four disease-associated gene sets as compared to both the soCRC and to the 1,000 Genomes cohorts (FDR <5%, Fig. 4b; Supplementary Data 8). Among these we found

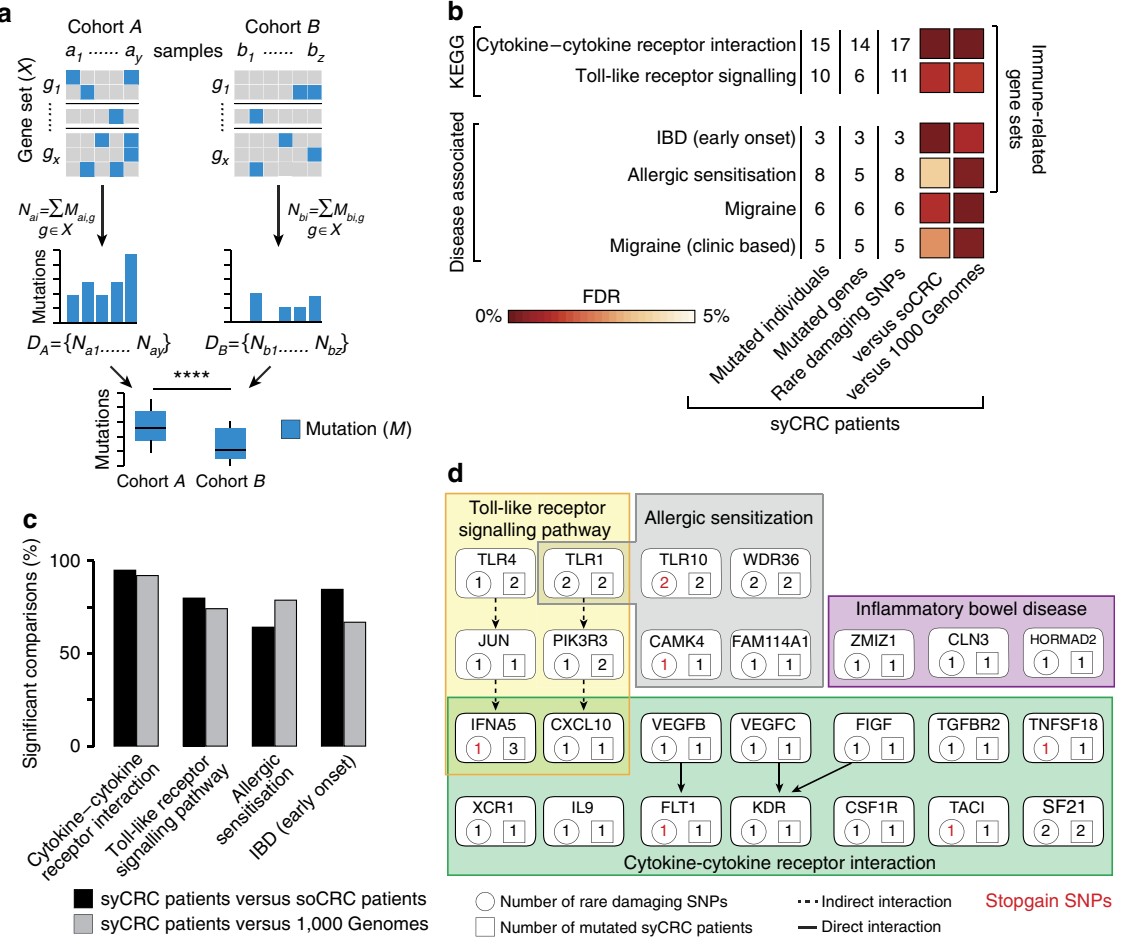

**Figure 4 | Rare damaging SNPs in immune-related genes of syCRC patients.** (**a**) Mutation enrichment gene set analysis (MEGA). All mutations in members of a given gene set are counted in all samples of two cohorts. The resulting distributions of mutations are compared between the two cohorts using the one-tailed Wilcoxon rank-sum test and corrected for multiple testing using the Benjamini & Hochberg method. (**b**) KEGG and disease-associated gene sets that are enriched in rare damaging SNPs in syCRC patients (33) when compared to soCRC patients (406) and to individuals from the 1,000 Genomes Project (756). Only gene sets with FDR <10% in both comparisons are reported. Since FDR was always <5%, FDR ranges from 0 to 5%. (**c**) Results of the bootstrapping procedure applied to the four immune-related gene sets. Significant comparisons had P value <0.05, one-tailed Wilcoxon rank-sum test. (**d**) Genes of the four immune-related gene sets with rare damaging SNPs in syCRC patients.

IBD, which is known to predispose to syCRC[10–12] and four of the enriched gene sets were clearly related to immune response (Fig. 4b). We also controlled for the effect of sample size (33 syCRC patients as compared to 406 soCRC patients and to 756 individuals from the 1,000 Genomes) using a bootstrapping procedure. For 10,000 times, we randomly extracted 33 individuals from the soCRC and from the 1,000 Genomes cohorts separately. At each iteration, we compared the distribution of rare damaging SNPs in the four immune-related gene sets between the 33 syCRC patients and the randomly extracted individuals. We found that syCRC patients had significantly more rare damaging SNPs (P value <0.05, one-tailed Wilcoxon rank-sum test) in the vast majority of comparisons (Fig. 4c).

Several gene sets that were frequently mutated in syCRC patients are involved in immune response. The cytokine-cytokine receptor interaction and the Toll-like receptor signalling pathways mediate immune response while IBD and allergic senitization arise in response to abnormal antigen recognition. Overall, 24 out of 33 syCRC patients (73%) had rare damaging SNPs in these four immune-related gene sets (Fig. 4d), with an average of one mutation per patient. As a comparison, only 94 soCRC patients (23%) and 200 individuals from 1,000 Genomes

(26%) had rare damaging SNPs in the same gene sets, with an average of 0.3 mutations per individual (Supplementary Data 6 and 8). Most of the genes in these gene sets were mutated in only one syCRC patient (Fig. 4d), confirming that it is the alteration of the immune response rather than of a specific gene to recur across these patients. The only exception was the stopgain mutation in the interferon domain of *IFNA5* that recurred in three patients (Fig. 4d; Supplementary Data 5).

**syCRC patients have abnormal mucosa immune composition.** To understand whether inherited alterations of immune-related genes were reflected in differences in the immune cell composition, we stained and counted T cells in the normal colonic mucosa of syCRC and soCRC patients. We found significantly higher fraction of CD8+ T cells in the normal mucosa, which was also evident when the lamina propria was scored independently (P values = 0.03 and 0.028, respectively, one-tailed Wilcoxon rank-sum test, Fig. 5a,b). Moreover, intraepithelial T cells were more frequent in the surface epithelium of syCRC patients, though not in the crypt epithelium (P values = $9 \times 10^{-3}$ and 0.5, respectively, one-tailed Wilcoxon rank-sum test, Fig. 5c,d). Therefore, the normal colonic mucosa of syCRC

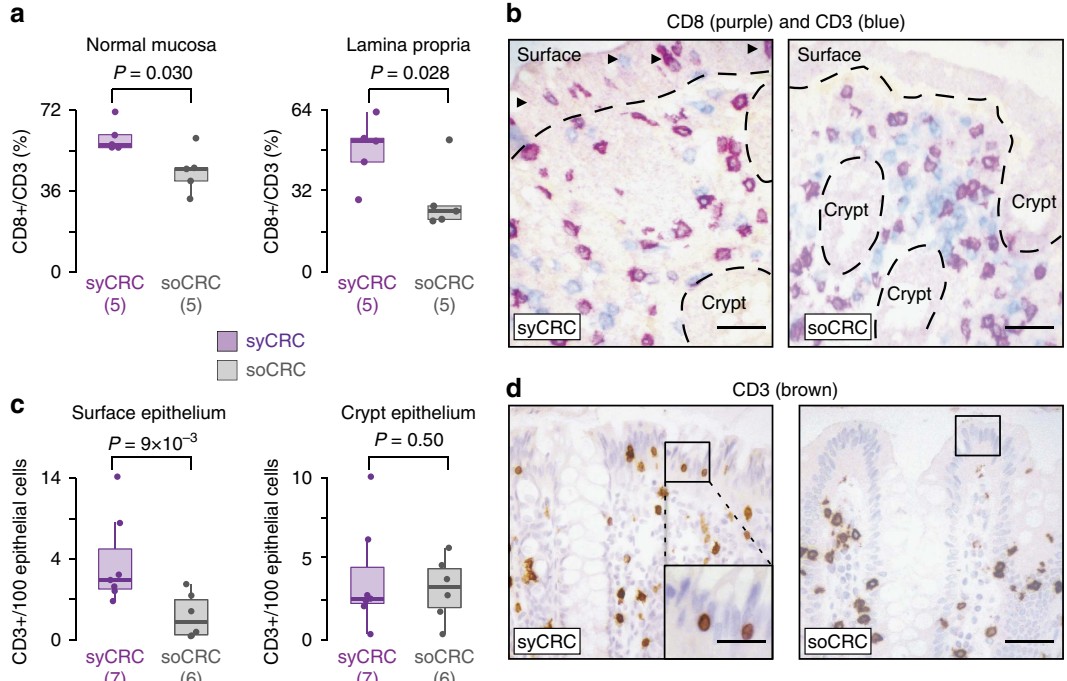

**Figure 5 | Immune cell composition of normal colonic mucosa of syCRC and soCRC patients.** (**a**) Percentages of CD8+ T cells over all CD3+ cells in the normal mucosa and lamina propria of syCRC and soCRC patients. Distributions were compared using the one-tailed Wilcoxon rank-sum test. (**b**) Example of immunostaining for CD8 (Novocastra, Clone 4B11) and CD3 (Novocastra, Clone LN10). Dotted lines mark the boundary between the epithelium and lamina propria. Arrowheads indicate intraepithelial lymphocytes in the surface epithelium. Magnification 200×; scale bar, 30 μm. (**c**) Percentages of CD3+ T cells in the surface epithelium and colon crypts of syCRC and soCRC patients. The number of samples is different in the two comparisons because not all types of information were available for all samples. Distributions were compared using the one-tailed Wilcoxon rank-sum test. (**d**) Example of immunostaining for CD3 cells with a blue nuclear counterstain. A representative area used to score frequency of surface intraepithelial lymphocytes is boxed and included as a magnified insert in the case of syCRC. Original magnification, 200×; scale bar, 30 μm. For each distribution, reported are the median value (horizontal line) and 1.5 times the interquartile ranges (whiskers).

patients has a different immune cell composition than the normal colonic mucosa of soCRC patients. To verify whether differences were also detectable in the tumours, we compared the levels of immune cell infiltrates in syCRCs and soCRCs. We measured the neutrophil-to-lymphocyte ratio because it has a prognostic value in colorectal cancer[39,40] and found that it was significantly higher in syCRCs ($P$ value $= 1.5 \times 10^{-4}$, one-tailed Wilcoxon rank-sum test, Fig. 6a). This was due to a higher number of neutrophils in syCRC (Supplementary Fig. 12). Since microsatellite instable (MSI) tumours show higher immune cell infiltrates, usually lymphocytes[41], we removed MSI tumours and verified that the difference between syCRCs and soCRCs remained significant ($P$ value $= 1.5 \times 10^{-3}$, one-tailed Wilcoxon rank-sum test, Fig. 6b). We then analysed only syCRCs and verified that the neutrophil-to-lymphocyte ratio was similar in MSI and microsatellite stable (MSS) syCRCs and slightly higher in T2 than in T1 ($P$ values $= 0.69$ and 0.013, respectively, one-tailed Wilcoxon rank-sum test, Fig. 6c). Thus, despite being genetically heterogeneous, syCRCs showed consistently higher level of tumour-associated inflammation, particularly in neutrophils (Fig. 6d). Finally, we sought to investigate whether these differences in the immune cell composition were reflected at the transcriptional level. We compared the entire transcriptomes of syCRCs and soCRCs using an approach conceptually similar to MEGA. First, we derived the overall distributions of gene expression in 14 syCRCs and 193 soCRCs that had RNA sequencing data in TCGA. Then, we grouped genes in four classes (not expressed, lowly expressed, medium expressed, highly expressed; see Methods section). Finally, we compared the fractions of genes in each of the four classes between syCRCs

and soCRCs in the 186 KEGG gene sets to search for significant differences. With the exception of the olfactory transduction pathway, we found no difference in the fractions of highly and medium expressed genes (Supplementary Data 9). Instead, syCRCs showed significantly higher fractions of not expressed and lowly expressed genes in 10 gene sets, seven of which had clear connections with immune response (FDR < 10%, Fig. 6e; Supplementary Data 9). In particular, the cytokine–cytokine receptor interaction and the Toll-like receptor signalling pathways had an average of 10 and 3% not expressed genes per tumour, respectively (Fig. 6e; Supplementary Data 9). We confirmed similar results with the bootstrapping to control for sample size (Supplementary Fig. 13) and with a gene set enrichment analysis of differentially expressed genes. In this case, we found 47 significantly down regulated gene sets (FDR < 9%), 18 of which are immune-related including all those found in the previous analysis (Supplementary Data 10). Finally, we observed an overall tendency of genes in the immune-related pathways (Figs 4d and 6e) to have lower expression when mutated as compared to wild-type, particularly in the presence of stopgain mutations (Supplementary Fig. 14). Altogether, these results indicated a different functionality of immune-related processes at the transcriptional level in syCRCs as compared to soCRCs.

## Discussion

In this study we show that syCRCs have independent genetic origins and develop into genetically heterogeneous tumours, in agreement with single observations in other anatomical sites such as kidney[42] and lung[43]. In general, tumour heterogeneity affects

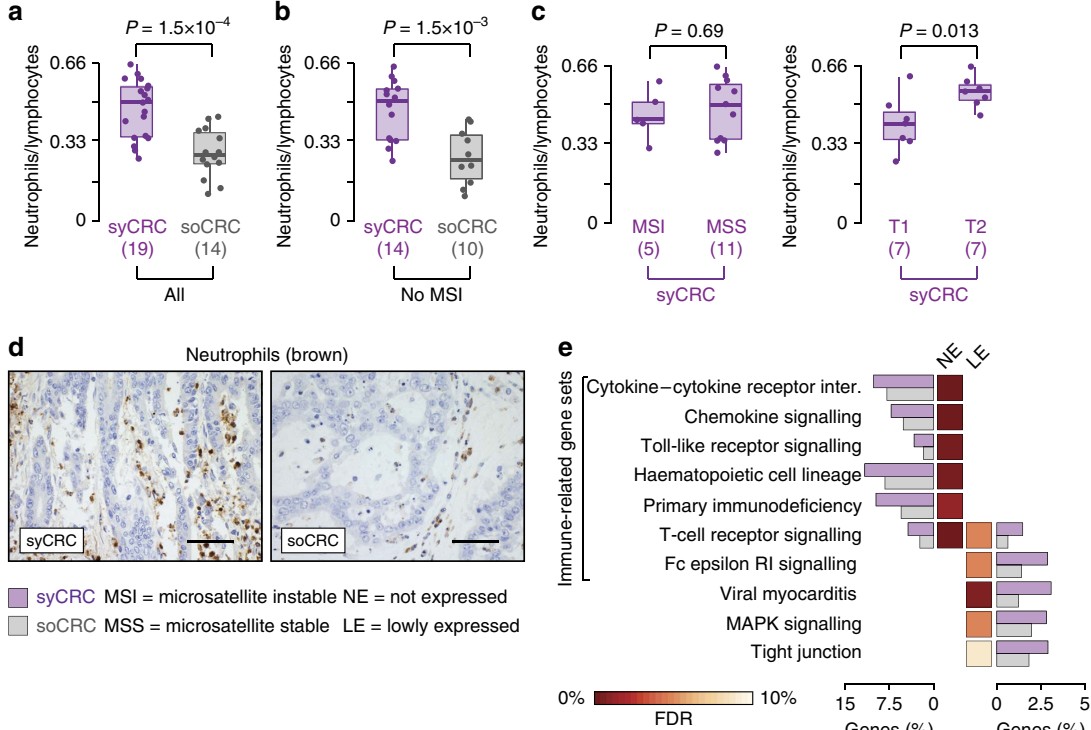

**Figure 6 | Immune cell composition and transcriptional differences in syCRCs and soCRCs.** (**a**) Neutrophil-to-lymphocyte ratio in syCRCs and in soCRCs. The number of samples is different in the two comparisons because not all types of information were available for all samples. Distributions were compared using the one-tailed Wilcoxon rank-sum test. (**b**) Neutrophil-to-lymphocyte ratio in MSS syCRCs and in MSS soCRCs. Distributions were compared using the one-tailed Wilcoxon rank-sum test. (**c**) Neutrophil-to-lymphocyte ratio in MSI and MSS syCRCs and in lesion T1 and T2 of syCRC patients. Distributions were compared using the one-tailed Wilcoxon rank-sum test. (**d**) Examples of immunostaining for neutrophils (neutrophil elastase) in syCRC and soCRC. Magnification 400 × ; scale bar, 30 μm. (**e**) KEGG gene sets that were enriched (FDR <10%) is not expressed and lowly expressed genes in syCRCs (14) as compared to soCRCs (193). Average percentage of genes in each gene set in the two cohorts of tumours is shown. For each distribution, reported are the median value (horizontal line) and 1.5 times the interquartile ranges (whiskers).

response to treatment because it may reduce the efficacy of therapies targeting specific gene aberrations[44]. In the particular case of syCRC, the scenario is complicated by the presence of genetically distinct tumours. Currently, no specific guidelines exist for the management of syCRC[45]. Our results highlight the need for testing all synchronous tumours to inform on the most appropriate clinical decisions. For example, anti-EGFR therapy and monitoring of mutations in EGFR downstream effectors as biomarkers of resistance may have limited applicability in syCRC patients because of the genetic contribution of distinct clones from multiple tumours.

We also contribute a better understanding of the factors that predispose to multiple tumours. Our analysis of germline mutations suggests that inherited damaging alterations of immune-related genes may increase the frequency of independent cancer-initiating events. This may be due to the inflammatory microenvironment that favours tumorigenesis via increased genomic instability or through the production of cytokines and growth factors[46,47]. In this respect, our results suggest the presence of an environmental field effect mediated by inflammation promoting the onset of multiple independent tumours. In addition to germline alterations, we report a higher proportion of CD8 + T cells in the lamina propria and a higher abundance in the surface intraepithelial compartment of the normal colon mucosa of syCRC patients. CD8 + T cells, in particular those with intraepithelial location, are associated with maintenance of epithelial integrity[48]. Thus, it is not clear if their relative abundance in syCRC is associated with tumorigenesis or increased surveillance. We also observe a

higher neutrophil-to-lymphocyte ratio and differences in the tumour expression of immune related genes. The role of immune cells in promoting neoplastic progression is well known[46,49], and it is usually mediated through the production of cytokines[46,50]. Moreover, bacterial biofilms, which contribute to chronic inflammation, are also known to favour the onset of colorectal cancer[51,52]. It would be interesting to check whether the gastrointestinal microbiome also contributes to the environmental field effect of syCRC. Further studies on the deregulation mechanisms of these immune processes are required to fully understand the onset of multiple primary tumours, including extending the analysis to other organs.

## Methods

**Sample description.** Tumours used in this study were collected from patients diagnosed with CRC who underwent surgical resection of two syCRC in a window of time of 6 months maximum[8]. Samples were obtained from four (S3, S6, S12 and S13) and six (UH1, UH2, UH5, UH6, UH8 and UH11) patients from the Istituto Clinico Humanitas (ICH), Rozzano, Milan, Italy and the University College London Hospital (UCLH), London, United Kingdom, respectively. All patients provided written informed consent and the study followed the approved institutional guidelines (ICH: ICH-25-09, 07/05/2009, UCLH: 07/Q1604/17 and 11/LO/1613).

BAT-26, BAT-25, NR-21, NR-24 and MONO-27 mononucleotides were analysed by capillary gel electrophoresis and used as molecular markers of microsatellite instability in tumours of patients S3, S6, S12 and S13. A tumour was classified as microsatellite unstable when at least one marker was found altered. All tumours, except those from patients S3 and S12, were further screened for the lack of mismatch repair proteins. For the tumours of patients S13 and S6, nuclear expression of hMLH1 (clone G-168–15, 1:200, BD Biosciences) and hMSH2 (clone FE11, Calbiochem, 1:100, Merck Millipore) were investigated via immunohistochemistry. The lack of expression of hMLH1/hMSH2 was assessed

in all samples under an optical microscope independently by two histologists. For lesions of patients UH1, UH2, UH5, UH6, UH8 and UH11 the expression of hMLH1 (Clone G168-15, 1:200 Biocare Medical), hMSH2 (Clone 25D12, 1:100, Novocastra), hMSH6 (Clone 44, 1:400, BD Biosciences Pharmingen) and hPMS2 (Clone A16-4, 1:300 BD Biosciences Pharmingen) were assessed using the Leica Vision BioSystems Bond-max.

**Quantification of tumour content.** The tumour content of lesions from patients S3, S6, S12 and S13 was measured as the average of three 2 μm-thick sections at the beginning, in the middle, and at the end of the region used for DNA extraction. Two independent pathologists quantified tissue and tumour areas using the ImageJ software (http://imagej.nih.gov/ij/) on the digitalized image of each section. The tumour content of the section was calculated as the percentage of tumour area over the total tissue area. The tumour area of lesions from patients UH1, UH2, UH5, UH6, UH8 and UH11 was delimited by the pathologist on haematoxylin and eosin stained 2 μm-thick FFPE sections at the beginning of the block. This section was then used to as a reference to macrodissect the tumour in each section. According to pathologist evaluation, the tumour content of the dissected areas was >90%.

**DNA extraction and whole-exome sequencing.** Genomic DNA for all tumours, except S3T1, and 7-matched normal tissues (S6N, UH1N, UH2N, UH5N, UH6N, UH8N, UH11N) was extracted from 10 μm-thick FFPE sections (3–6 sections per sample) using QIAamp DNA FFPE Tissue kit (Qiagen) following the manufacturer's protocol. Tumour S3T1 derived from fresh frozen tissue. Blood of patients S3, S12 and S13 was used as matching reference. DNA from blood and frozen samples was extracted using DNaesy Blood & Tissue kit (Qiagen) according to the manufacturer's protocol.

Whole exome was captured from genomic DNA for all 20 tumours and matched normal using the SureSelect XT Human All Exon V4 (Agilent) following the manufacturer's protocol with modifications in case of DNA extracted from FFPE samples. Briefly, 3 μg of genomic DNA was sheared using an Adaptive Focused Acoustics technology (Covaris) to obtain ~200-bp-long fragments. Fragments were used to prepare libraries according to SureSelect XT manual. Libraries were further amplified with 7–10 cycles of PCR and 500–750 ng were hybridized with the bait library. Captured DNA was amplified with 16 PCR cycles and barcode indexes were added. Libraries of tumour and normal samples from patients UH8 and UH11 were prepared from 500 ng of genomic DNA using NEBNext Ultra DNA Library Prep Kit for Illumina (NEB) with minor modifications to make it compatible with SureSelect XT Human All Exon V4 capture kit (Agilent). Briefly, NEB adaptors and NEB PCR primers were replaced by SureSelect adaptor mix and SureSelect ILM indexing pre capture primers, respectively. Libraries were then sequenced using one lane (S13N, S13T1, S13T2, S6N, S6T1, S6T2, S3N, S3T1, S3T2, S12N, S12T1, S12T2) or half a lane (UH1N, UH1T1, UH1T2, UH2N, UH2T1, UH2T2, UH5N, UH5T1, UH5T2, UH6N, UH6T1, UH6T2) of Illumina HiSeq 2000 or one-third of a lane of Illumina HiSeq 2500 (UH8N, UH8T1, UH8T2, UH11N, UH11T1, UH11T2) per sample, with 76 and 101 bp paired-end protocol, respectively. All tumour and normal samples of patients S3, S6, S12 and S13 underwent a second round of whole-exome sequencing from independent libraries. DNA extraction and library preparation were performed as described above. Each library was next sequenced using half a lane of Illumina HiSeq 2000 with 76 bp paired-end protocol. Samples were sequenced at the sequencing facility of the IFOM-IEO Campus, Milan and at the Biomedical Research Centre Genomics Core Facility, Guy's Hospital, London.

**Sequence alignment and variant annotation.** Sequencing reads from each sample were aligned to the human genome (GRCh37/hg19) using Novoalign (http://www.novocraft.com/) with default parameters. At the most three mismatches per read were allowed and PCR duplicates were removed using rmdup of SAMtools[53]. All reads uniquely mapping within 75 or 100 bp from the targeted regions were considered as on target and retained for further analysis. SNVs and small insertion/deletions (InDels) were identified using VarScan2 (ref. 54) in each tumour and in normal samples independently. In tumours, SNVs and InDels were further retained if (1) supported by at least 10 mutated reads, (2) had allele frequency ≥5%, and (3) had at least 1% of reads mapping on both DNA strands. In normal samples, SNVs and InDels were further retained if (1) supported by at least two mutated reads and (2) had at least 1% of reads mapping on both DNA strands. For normal and tumour samples that underwent two whole-exome sequencing rounds, mutations were called independently in each experiment as described above. In the case of normal samples, mutations called in each round were merged and used for further analysis. In the case of tumour samples, SNVs and InDels detected in one round were retained only if present in the other sequencing round and merged. In each tumour, somatic SNVs and InDels were identified as tumour-specific if absent in the normal counterpart and further retained after manual inspection. MuTect[55] (version 1.17) and Strelka[56] (version 1.0.14) were used to measure concordance in calling SNVs and InDels, respectively, with default parameters (MuTect: minimum number reads covering a site in the tumour = 14 and in the normal = 8; Strelka: indelMaxRefRepeat = 8, indelMaxWindowFilteredBasecallFrac = 0.3, indelMaxIntHpolLength = 14, sindelPrior = 0.000001, sindelNoise = 0.000001, sindelQuality_LowerBound = 15)

to all tumours and matched normal samples. Only somatic SNVs identified as 'KEEP' in MuTect and InDels identified as 'PASS' in Strelka were retained and intersected with the manually curated collection of somatic mutations in each sample.

Starting from the entire pool of somatic mutations in each tumour, ANNOVAR[57] was used to identify nonsilent (nonsynonymous, stopgain, stoploss, frameshift, nonframeshift and splicing modifications) mutations using RefSeq v.64 (ref. 58) as a reference protein dataset. SNVs and InDels falling within 2 bp from the splice sites of a gene in one of the three datasets were considered as splicing mutations.

Thirty-five somatic nonsilent mutations were randomly selected from all samples for orthogonal validation. Genomic regions of ~200 bp long encompassing the mutations were amplified by PCR using the Q5 High Fidelity DNA Polymerase (New England Biolabs) and PCR amplicons were submitted for Sanger sequencing. Chromatograms were processed with Chromas 2.3 and all sequences were visually inspected.

**Deep sequencing of cancer gene panel.** Genomic DNA was extracted from macrodissected tumours UH1T1, UH1T2, UH2T1, UH2T2, UH11T1 and UH11T2, and libraries were prepared as described above. A panel of 151 cancer genes was then captured in each tumour using ClearSeq Comprehensive Capture kit (Agilent) and sequenced in one lane of Illumina Miseq using the 300 paired-end protocol. Samples were sequenced at the sequencing facility of the Biomedical Research Centre Genomics Core Facility, Guy's Hospital, London. Alignment of sequencing reads and variant calling were performed using the same analytical framework as described above.

**SNP array and copy number detection.** Quality of the genomic DNA extracted from FFPE blocks was assessed using Infinium HD FFPE QC kit (Illumina) and DNA was restored using Infinium HD FFPE restore kit (Illumina). Tumour and matched-normal samples were genotyped using HumanOmniExpress-24 v1.0 (Illumina) and images were scanned using a BeadArray reader. Intensity and genotype data were extracted for CNV analysis after normalizing raw fluorescent signals using Illumina Genome Studio v2011.1.CNVs were detected using ASCAT[59] (version 2.1) with default parameters (segment lengths for ASPCF Segmentation and probes with minor allele frequency ranging between 40 and 60% in matched reference for all samples). To improve the identification of CNVs with FFPE samples only high-quality-genotyped probes (genocall score >0.7) were used. Analysis of all tumour samples was done in comparison with matched normal. Frequency distributions of the germline heterozygous single SNPs were integrated with the SNP array results to identify high-confidence aberrant regions. In a diploid genome, heterozygous SNPs follow a normal distribution centred around 50% allele because both alleles are present at equal frequency. In the case of allelic imbalance due to CNVs, frequency distribution of heterozygous SNPs deviates from normality because of the unbalanced ratio between mutated and wild-type alleles. Hence, the distribution of heterozygous SNP frequencies can be used to identify genomic regions undergoing CNVs. High-confidence aberrant regions were defined as genomic segments with copy number different from 2 and with an aberration reliability score >75%, and present in regions with non-normal SNP frequency distribution. The copy numbers of aberrant regions were assessed using ASCAT[59]. To identify amplified and deleted genes, the genomic coordinates of the aberrant regions in each sample were intersected with those of 21,033 human genes of the SureSelect XT Human All Exon V4 kit (Agilent). A gene was considered as modified if ≥80% of its length was contained in an aberrant region[60].

**Identification of putative driver alterations.** A list of CRC genes was retrieved from the Network of Cancer Genes[61](http://ncg.kcl.ac.uk/). This list was intersected with the list of genes with somatic mutations, amplifications and deletions in each tumour. Altered cancer genes were further classified as putative drivers in each tumour if they harboured (1) nonsilent mutations and/or (2) were oncogenes undergoing amplification or (3) tumour-suppressors undergoing deletion in recurrently modified CRC regions (http://www.broadinstitute.org/tcga/home, version '2014-11-03').

**Reconstruction of tumour clone composition.** Clone composition of each tumour was rebuilt based on the clonality of its somatic alterations (SNVs, InDels, amplifications and deletions), defined as the fraction of cancer cells carrying each alteration. All CNV regions and all SNVs and InDels were used, except mutations falling in amplifications and deletions, because of the effect of CNVs on allele frequency.

Clonality of SNVs and InDels was assessed based on the allele frequency (number of mutated sequencing reads over the total number of reads covering that position). First, the allele frequency ($f$) of each mutation ($i$) was corrected for the tumour content (tc):

$$f'_i = \frac{f_i}{tc} \qquad (1)$$

Second, the clonality ($c$) of each mutation ($i$) was measured as the double of the corrected allelic frequency ($f'$), to account for the presence of two alleles per cell:

$$c_i = f'_i \times 2 \qquad (2)$$

In cases of mutations with $50\% < f' < 100\%$ (almost clonal), clonality was assessed as:

$$f'_i > 50\% \rightarrow \begin{cases} c'_i = 100\% \\ c''_i = \left(f' - 50\%\right) \times 2 \end{cases} \quad (3)$$

where $c'_i$ and $c''_i$ represent the clonality of the two alleles, respectively.

Clonality of regions undergoing amplification or deletion was assessed based on the variation in allele frequency of heterozygous SNPs between each tumour and the matched-normal sample. First, heterozygous SNPs in each somatic CNV were identified as germline mutations with 40–60% allele frequency in the matched-normal tissue of each patient. Second, the allele frequency of any CNV region ($j$) was measured as the median allele frequency ($\bar{f}$) of all heterozygous SNPs in the region and corrected for the tumour content:

$$\tilde{f}'_j = \frac{\tilde{f}_j}{\text{tc}} \quad (4)$$

Third, the clonality ($c$) of each CNV region ($j$) was measured as twice the absolute deviation of $\tilde{f}'$ from the expected allele frequency (50%), to account for CNV allelic imbalance:

$$c_j = \left| \tilde{f}'_j - 50\% \right| \times 2 \quad (5)$$

The density distribution of clonality was calculated for each type of alteration independently using the one-dimensional Gaussian kernel estimator as implemented in the R function 'stat_density' (http://ggplot2.org).

First, the probability density ($\rho$) was derived from the clonality of all somatic alterations. It defined the probability ($P$) of observing an alteration ($x$) at clonality ($I$) as the area of density distribution in $I$:

$$P(x) = \int_I \rho(x)\,\mathrm{d}x \quad (6)$$

Second, the expected number of alterations ($E$) at clonality ($I$) was calculated as the probability of each alteration $P(x)$ multiplied by the total number of alterations ($n$)[62]:

$$E = P(x) \times n \quad (7)$$

The density distribution of clonality corresponded to the distribution of the expected number of alterations and recapitulated the clonal composition of the tumour.

**Identification of rare damaging SNPs.** The BAM files of matched-normal tissue of 429 TCGA CRC samples (304 colon and 125 rectum adenocarcinoma) were downloaded from the Cancer Genomics Hub (https://cghub.ucsc.edu). The corresponding clinical data were downloaded from the TCGA data portal (https://tcga-data.nci.nih.gov/tcga/dataAccessMatrix.htm). This information was used to identify 23 syCRC and 406 soCRC patients and to assess the occurrence of extra-colonic malignancies. TCGA patients were classified as affected by hereditary cancer based on the immunohistochemistry of mismatch-repair proteins, on the microsatellite instability, or on the presence of pathogenic SNP in CRC predisposing genes. The BAM files of 756 non-consanguineous individuals from the 1,000 Genomes Project[37] were downloaded from the repository (ftp://ftp.1000genomes.ebi.ac.uk, phase 1 version 3 April 2014). All samples had sequencing coverage >20 reads on at least 70% of the targeted regions.

To obtain uniform variant calling across samples, all BAM files from 10 syCRCs sequenced here, 23 TGCA syCRCs, 406 TCGA soCRCs and 756 individuals from the 1,000 Genomes Project were re-analysed with the same pipeline to identify germline SNPs. Briefly, PCR duplicates were removed using rmdup command of SAMtools[53] and only reads uniquely mapping within the SureSelect XT Human All Exon V4 kit targeted regions were used for further analysis. SNPs were identified using VarScan2 (ref. 54) and retained if (1) supported by at least 10 mutated reads and 1% of the reads mapping on both DNA strands, and (2) had allele frequency ranging between 40 and 60% or >90%. For samples S3, S6, S12 and S13 that underwent two rounds of sequencing, aligned BAM files of the normal tissue were merged using SAMtools[4] before variant calling. ANNOVAR[57] was used to identify stopgain, stoploss and nonsynonymous mutations using RefSeq v.64 (ref. 58) as protein dataset. All stopgain and stoploss SNPs were considered as damaging, while predictions on the damaging effect of nonsynonymous mutations were obtained from dbNSFP version 2.4 (ref. 63), based on SIFT[64], PolyPhen2[65], MutationTester[66], MutationAssessor[67] and LTR[68]. Mutations were considered as damaging if predicted by at least four of the five methods (SIFT score <0.05, labelled as 'probably damaging' or 'possibly damaging' by PolyPhen2, MutationTester and LTR or as 'high predicted non-functional' by MutationAssessor). Rare damaging SNPs were defined as those with either no minor allele frequency or with minor allele frequency lower than 1%, as reported in ANNOVAR based on the 1,000 Genomes Project (phase 1 version 3, April 2014) and on the NHLBI GO Exome Sequencing Project (version 0.0.27, April 2014). To further remove possible sequencing and alignment errors, all rare damaging SNPs not detected in the general population but present in more than 50% of the three cohorts were removed.

**Mutation enrichment gene set analysis (MEGA).** MEGA was developed to identify gene sets (for example, genes involved in the same pathway, or predisposing to specific diseases) that show a significantly higher number of mutations in syCRC patients as compared to soCRC patients.

As an input, MEGA requires a gene set $X = \{g_1, \ldots, g_x\}$ and a list of mutations detected in this set in two cohorts of samples $A = \{a_1, \ldots, a_y\}$ and $B = \{b_1, \ldots, b_z\}$. For each sample ($a_i, b_i$) in cohorts $A$ and $B$, the number of mutations ($N_{a_i}, N_{b_i}$) in all genes of gene set $X$ is calculated as:

$$N_{a_i} = \sum_{g \in X} M_{a_i,g} \text{ and } N_{b_i} = \sum_{g \in X} M_{b_i,g} \quad (8)$$

where $M_{a_i,g}$ and $M_{b_i,g}$ are all mutations in gene $g$ for samples $a_i$ and $b_i$, respectively.

The distributions of mutations ($D_A$ and $D_B$) for cohorts $A$ and $B$ are then derived as:

$$D_A = \{N_{a_i}, \ldots, N_{a_y}\} \text{ and } D_B = \{N_{b_i}, \ldots, N_{b_z}\} \quad (9)$$

To determine whether cohort $A$ is enriched in mutations of gene set $X$ as compared to cohort $B$, distributions $D_A$ and $D_B$ are compared using the one-tailed Wilcoxon rank-sum test. In the case of multiple gene sets, the $P$ value from each comparison is corrected for FDR using the Benjamini & Hochberg method. In case the sample sizes differ substantially between groups $A$ and $B$ (as in the case of our cohorts), MEGA applies a bootstrapping procedure (random sampling with replacement). It down-samples the larger cohort to reach the sample size of the smaller cohort randomly 10,000 times and repeats the analysis at each iteration. At the end of all iterations, MEGA calculates the proportion of significant enrichments ($P$ value <0.05, one-tailed Wilcoxon rank-sum test) over the total comparisons.

Two lists of gene sets were used here. The first list was composed of 186 manually curated gene sets comprising 5,267 human genes from the Kyoto Encyclopedia of Genes and Genomes (KEGG) were downloaded from MSigDb[35] (version 5). The second list of 1,076 diseases was collected from the catalogue of published Genome-Wide Association Studies (GWAS, October 2013)[38]. Genes associated with diseases were retrieved and grouped into 1,076 disease-associated gene sets. Only 346 of these (comprising 6,589 human genes) were selected because they contained at least 10 genes. For each gene set the number of rare damaging SNPs were compared between syCRC and soCRC patients or individuals from 1,000 Genomes Project.

**Immune composition of tumour and normal mucosa.** Immunohistochemistry was performed to identify CD8 (Clone 4B11, 1:1, Novocastra) and CD3 (Clone LN10, 1:1, Novocastra) double immunostaining heat mediated antigen retrieval was performed using Bond Epitope Retrieval Solution 1 (pH 6.0, citrate) for 30 min before the tissue was blocked using peroxide for five minutes. After primary antibody incubation for 15 min, eight minute incubation of both post primary reagent and a polymer reagent was performed. Bond polymer refine red detection was used for visualization of the first antibody. A second heat mediated retrieval using Bond epitope retrieval solution 2 (pH 9.0, TE) was used for the second antibody for 10 min, followed by incubations of 30 min for the second antibody, and 20 and 30 min for the post primary and polymer reagents, respectively. DAB was used for visualization. Haematoxylin was used as a counterstain. The density of the intraepithelial T-cell compartment and the ratios of CD8+ to CD8− T cells were determined using our published methods[69,70]. To identify neutrophils (Neutrophil elastase, Dako, clone NP57) and lymphocytes (CD3, Novocastra, clone LN10) 19 syCRCs (from Instituto Humanitas Milan, UCLH London, and Wellcome Trust Centre for Human Genetics, Oxford) and 14 soCRCs (from UCLH London) were stained on the bondmax system (http://www.vision-bio.com/). The number of neutrophils and lymphocytes was counted in three different areas (hot spots) away from the luminal surface of the tumour. The histopathologist was blinded to the tumour details (MSI, MSS, syCRC, soCRC) and areas of ulceration, necrosis, artefacts, and intra-vascular neutrophils were ignored. The tumour sections were screened at low power magnification ($\times 40$ and $\times 100$), and 2 high power fields ($\times 400$). The average number of infiltrates was expressed in number per high-power field (Olympus BX51, $\times 400$).

**Gene expression analysis.** RNA sequencing data (level 3, RNASeqV2) were available in TCGA for 14 syCRCs, 193 soCRCs and 35 matching normal samples. Starting from the scaled estimate expression values for 20,531 genes, the number of transcripts per million reads (TPM) was obtained. Gene expression in normal tissue was calculated as the average TPM across the 35 normal samples. The distribution of TPM values was measured in each sample and genes were considered as (1) highly expressed, if their TPMs fell in >75th percentile of the distribution; (2) medium expressed, if their TPMs ranged between 25th and 75th percentile of the distribution; (3) lowly expressed, if their TPMs fell in <25th percentile of the distribution; (4) not expressed in the tumour, if their TPMs were <0.1. The cumulative proportion of genes in each class of expression levels was compared between syCRCs and soCRCs using the Fisher's exact test in each of the 186 KEGG gene sets. The resulting $P$ values were corrected for multiple tests using the Benjamini & Hochberg method. To control for the sample size effect, a bootstrapping procedure was applied. For 10,000 times, 14 soCRCs were randomly selected and the proportion of genes in the four classes of expression in each gene set was compared with that of 14 syCRCs using the Fisher's exact

test. At the end of all iterations, the proportion of significant enrichments ($P$ value $< 0.05$, Fisher's exact test) over the total comparisons was calculated.

Differentially expressed gene sets between 14 syCRCs and 193 soCRCs were detected using the GAGE package (http://www.bioconductor.org/packages/release/bioc/html/gage.html). Starting from the read counts of 20,531 genes in each samples, genes with read count equal to zero across all samples were removed. For each sample the number of reads was normalized to the total amount of sequenced reads and a $\log_2$ transformation was applied to stabilize the variance at low expression levels. GAGE was applied with default parameters (set.size = c(10,500); rank.test = FALSE; use.fold = TRUE; FDR.adj = TRUE; weights = NULL; saaPrep = gagePrep; saaTest = gs.tTest; saaSum = gageSum; use.stouffer = TRUE) and experimental design specified as 'unpaired'. A FDR threshold of $< 10\%$ was used to detect down and up regulated gene sets in syCRCs as compared to syCRCs.

**Code availability.** MEGA is implemented under the R software environment (https://www.r-project.org). It is publicly available at https://github.com/ciccalab/MEGA.git and as Supplementary Software 1.

**Data availability.** Whole-exome sequencing and SNP array data for this study have been deposited in the European Genome-phenome Archive (EGA) under the accession number EGAS00001001461. TCGA data were downloaded from https://cghub.ucsc.edu and from https://tcga-data.nci.nih.gov/tcga/dataAccessMatrix.htm. Overall, 1,000 Genomes Project data were downloaded from ftp://ftp.1000genomes.ebi.ac.uk. All of the remaining data is available within the article and Supplementary Files or available from the author upon request.

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

## Acknowledgements

This work was supported by the Italian Association for Cancer Research (AIRC IG-12742 to F.D.C.), by the National Institute for Health Research (NIHR) Biomedical Research Centre based at Guy's and St Thomas' NHS Foundation Trust and King's College London (to F.D.C.) and by the UCLH/UCL NIHR Biomedical Research Centre (to D.P. and M.R.-J.). We thank Dr Claire Palles and Prof Ian Tomlinson (Wellcome Trust Centre for Human Genetics Oxford) for additional samples for immunostaining and Prof Peter Parker (King's College London) for discussions and comments on the manuscript. Open access for this article was funded by King's College London.

## Author contributions

F.D.C. conceived and directed the study; M.C., G.G. and F.I. developed the computational pipelines; L.B. and R.F.G. generated and validated sequencing data; M.C., G.G., L.B., F.D.C. and F.I. analysed the data; S.S. and F.I. analysed copy number alterations; M.C. and T.P.M. analysed gene expression; L.B. and I.P. macrodissected the samples; D.P., G.B., L.L. and M.R.-J. provided the samples; M.R.-J. and L.L. contributed pathological inspection; D.P. and M.R.-J. performed immunostanining and measured tumour infiltrates; J.S. analysed the immune cells in normal mucosa; F.D.C., M.C., G.G. and L.B. wrote the manuscript; F.I., L.L., J.S. and M.R.-J. edited the manuscript.

## Additional information

**Competing financial interest:** The authors declare no competing financial interests.

