## [Peer Review File · Nature Communications]

Reviewer #2 (Remarks to the Author):

In the revised manuscript by Cereda et al, the authors have responded to a number of concerns raised in the original text. In particular, aspects of the analysis and approach have been clarified and presented more coherently, the sample size has increased, and additional data/analyses have been introduced. Overall, the manuscript is significantly improved and there is greater support for the primary claims of the paper concerning the distinct genetic origins of syCRCs and the putative role of damaging germline variants in immune genes in their tumorigenesis. While many of the points have been addressed, there are a few minor outstanding issues that remain, as noted below.

-In order to evaluate evidence for the independent origin of syCRCs and whether shared cancer genes were likely to have been acquired at different stages of tumour growth in different lesions from the same patient (p.6), the "clonality" of a mutation was computed by adjusting the allele frequency (AF) for tumor content (purity). Based on this analysis, the authors conclude that in the majority of cases, "driver" genes had differing clonality. These clonality values are also utilized to compare overall clone composition and the status of "actionable" genes.

To ensure high tumor purity, the authors performed a detailed (and quantitative) pathology review, followed by macrodissection. This should be commended and likely contributed to improve data quality.

In addition, it is possible to molecularly infer tumor purity from the sequencing or SNP data and these values should correlate with the pathology estimates. For example, they can be obtained using ASCAT, which was employed for CN analysis. It would be informative to report these values, particularly for the lower purity samples.

Importantly, however, the accurate comparison of allele frequency (AF) estimates from two samples should account not only for tumor purity, but also for ploidy and regional copy number. To this end, it is standard to compute the cancer cell fraction or cellular prevalence of a mutation. Instead, the authors restrict their analyses of clonality to diploid regions of the genome. They note that 76% of mutations fall in diploid regions implying that little is missed by excluding non-diploid regions. This value of 76% seems surprisingly large number given the extent of aneuploidy in non MSI-high colon cancers. Indeed, Table S1 indicates that a large number of genes are altered in many cases (often with variability between the two tumors) and the few plots in Fig S6, suggest this could be extensive. Moreover, many colon cancers undergo whole genome doubling [PMID: 24436049] and this is not accounted for, but could skew the results influence the AF estimates depending on whether the mutation occurred prior to genome doubling. Since genome-wide copy number data is available, why not compute the CCF and perform a global analysis that accounts for ploidy/CN? At present, the spectrum of alterations identified in these tumors is not fully exploited and this has value in the inference of mutational timing, as relevant to Figure 2.

It is also worth noting that the cancer genes classified as "drivers" in the above analyses may function differently depending on the context in which they occur. To avoid confusion, these should be referred to as cancer genes.

-Returning to Table S1, why is it that some MSI-high tumors, which are typically diploid have such high numbers of copy number altered regions? For example Pt S6 (Lynch) has 3546 CNV genes in T1 and only 2 in T2. This is curious.

-The classification of alterations as clonal, biclonal, polyclonal is rather non-conventional although

there may be value in this simplified view. Nonetheless, it should be stated that neither subclonal copy number or clonal genotypes can be resolved from these data and that the density distributions are intended to provide a crude summarization of the expected number of alterations of a given class.

-The prediction of mutation pathogenicity is a challenging task and it is generally accepted that no single method has optimal performance across an array of functional categories or architectures. The authors employ a number of well established tools for this prediction, and take a union set approach by calling variants as damaging if they are categorized as such by 4/5 methods. This is a reasonable strategy, but it may be worth considering whether more recent methods such as CADD [PMID: 24487276] or DANN (a deep neural network version of CADD) [PMID: 25338716] both of which integrate diverse annotations identify additional high-confidence variants.

Reviewer #4 (Remarks to the Author):

A. Summary of the key results

The authors sought to understand the balance of genetic and environmental factors underlying synchronous colorectal cancers (syCRCs). Interestingly, this report found that syCRCs are genetically unrelated and highly heterogeneous. The authors also found:

1. inherited damaging mutations tended to occur in immune-related genes
2. different proportions of immune cell populations in tumour and normal mucosa

B. Originality and interest: if not novel, please give references

The paper is of interest to the field of immune-oncology. In particular, the finding that syCRCs are potentially completely distinct has implications for therapeutic strategies.

C. Data & methodology: validity of approach, quality of data, quality of presentation

The curation and processing of the genetic data was careful and presented well.

D. Appropriate use of statistics and treatment of uncertainties

Many of the statistical methods seemed ad hoc with assumptions about direction without a good rationale (e.g., one-sided Wilcoxon test) or convenient (stratifying gene expression rather than applying differential expression analysis). An FDR of 0.1 is reasonable, but also generous when a few gene sets are identified and the authors don't report the statistics for enrichments (e.g., OR and p-values). How do the authors deal with the uncertainty in estimating cell counts from slides?

E. Conclusions: robustness, validity, reliability

The genetic analysis was strong and the finding of heterogeneity is novel. The immune connection seems less reliable because of the statistical treatment and limited analysis. The authors set out to conduct a systematic exploration, yet restrict the gene sets to KEGG only, and examine only a couple immune cell types. Did the authors consider neutrophils and lymphocytes separately and not find differences? Also, the term constitutional alteration seems a bit strange and maybe too strong given the data. Also, I don't understand how the authors claim deregulation from the transcriptional data. The transcriptional levels are different, but it's not clear whether this is deregulation or some other mechanism.

F. Suggested improvements: experiments, data for possible revision

One hypothesis that wasn't explored, but would be interesting is what sort of environmental perturbation might trigger the cancer. For example, the authors could examine viral integration or the propensity of viral RNA in their samples to determine whether the germline mutations in immune-related genes predispose an individual to develop an oncogenic conversion following some sort of pathogenic or chemical perturbation that exploits the immune-related pathways (i.e., Toll). The presence of the immune cells are somewhat speculative and one cannot determine whether they are drivers or passengers. (my guess is the latter)

G. References: appropriate credit to previous work?

Yes

H. Clarity and context: lucidity of abstract/summary, appropriateness of abstract, introduction and conclusions

The paper is well organized and the arguments are clear. The authors need to fix some typographical and grammatical errors. For example, the abstract could be improved through some clear language and edits. I've made one change below.

"This inter- and intra-tumour heterogeneity has consequences for identifying effective treatments and monitoring resistance. To understand the causes of syCRCs, we searched for biological processes that are altered in syCRC patients compared to patients with solitary colorectal cancer or to healthy individuals."

Reviewer #2 (Remarks to the Author):

1. To ensure high tumor purity, the authors performed a detailed (and quantitative) pathology review, followed by macrodissection. This should be commended and likely contributed to improve data quality.

Response: We now comment on this (p. 4).

2. In addition, it is possible to molecularly infer tumor purity from the sequencing or SNP data and these values should correlate with the pathology estimates. For example, they can be obtained using ASCAT, which was employed for CN analysis. **It would be informative to report these values, particularly for the lower purity samples.**

Response: As suggested by the reviewer, we assessed tumour purity with ASCAT on all 20 tumours and measured the correlation with our estimations. Overall, we obtained a good correlation between the two measures except for four samples (S13T1, UH8T2 and UH11T2, S6T2, Panel A). When the four outliers were excluded, the correlation between the two estimations became stronger and highly significant (Panel B).

To assess what estimation of the two was more accurate in the four discordant cases, we relied on the highest allele frequency of somatic mutations in diploid regions in each tumour. This value can be lower (for subclonal and clonal heterozygous mutations) or equal (for clonal homozygous mutations) but not higher than the estimated tumour content.

In samples S13T1, UH8T2 and UH11T2, the highest frequencies of somatic mutations are 33%, 58%, 80%, respectively (Supplementary Table 3). These values are compatible with our estimations of tumour content (49%, 90%, 90%) but much higher than the tumour content from ASCAT (20%, 44%, 48%).

In sample S6T2, the highest frequency of somatic mutations is 24% (Supplementary Table 3).

Again, this is more compatible with clonal heterozygous mutations in a sample with a tumour content of 41% (our estimation) than of 90% (ASCAT).

Based on these comparisons, our estimation of tumour content from pathology review of macrodissected sections or across sections is more accurate than that inferred from ASCAT. Therefore, we decided not to report the results of ASCAT because they do not add any further support to the previous assessment.

The tendency of ASCAT to give inaccurate estimations under certain circumstances has been reported previously¹ and acknowledged by the same authors². Moreover, there is no standard way of assessing tumour purity. Some studies rely on the assessment from pathologists³⁻⁶, while others rely on computational approaches different from ASCAT (ie ABSOLUTE)^{1,7-14}.

*3. Importantly, however, the accurate comparison of allele frequency (AF) estimates from two samples should account not only for tumor purity, but also for **ploidy and regional copy number**. To this end, **it is standard to compute the cancer cell fraction or cellular prevalence of a mutation**. Instead, the authors **restrict their analyses of clonality to diploid regions of the genome**. They note that 76% of mutations fall in diploid regions implying that little is missed by excluding non-diploid regions. This value of 76% seems surprisingly large number given the extent of aneuploidy in non MSI-high colon cancers. Indeed, Table S1 indicates that a large number of genes are altered in many cases (often with variability between the two tumors) and the few plots in Fig S6, suggest this could be extensive. Moreover, many colon cancers undergo whole genome doubling [PMID: 24436049] and this is not accounted for, but could skew the results influence the AF estimates depending on whether the mutation occurred prior to genome doubling. **Since genome-wide copy number data is available, why not compute the CCF and perform a global analysis that accounts for ploidy/CN?** At present, the spectrum of alterations identified in these tumors is not fully exploited and this has value in the inference of mutational timing, as relevant to Figure 2.*

Response: Table S1 annotates the clinical features of samples; likely the reviewer refers to Table1. Fig.S6 shows an example of different alterations of the same cancer genes between paired lesions.

As explained in the previous point-by-point response, in the text (p.7, 30-31) and in Fig.2a, we did not restrict the analysis of clonality to diploid regions of the genome. Rather, we assessed the clonality of mutations **AND** of copy number variant regions separately to then combine both assessments and infer the clone composition. Therefore, the information that derives from regions of amplification and deletion is not ignored.

We got the value of 76% mutations that are not affected by CNVs by intersecting the coordinates of the alterations reported in Table S3 with the coordinates of copy number variant genes in Table S5. This number may be surprisingly large, but this is what we observed in our cohort of samples. Moreover, although whole genome duplication is common in CRC, we do not have any evidence that it occurs in our samples.

As requested by the reviewer and to further confirm that our approach is not introducing any bias in the analysis of clonality, we estimated the cancer cell fraction (CCF) of somatic mutations as previously described¹⁵. We then used these measures to calculate the clone composition of the 20 tumours. It is worth noting that CCF relies only on mutation data (SNVs and small indels) to

reconstruct the clone composition and does not consider the clonality of CNV regions unless they harbour somatic mutations. Our method, instead, uses both types of information.

Overall the clonality estimations were remarkably similar between the two methods resulting in identical clone composition for 17 out of 20 tumours (Supplementary Fig. 9).

We analysed in detail the three samples where the clone composition was different (S6T2, UH5T2 and UH8T2, Supplementary Fig. 9).

In samples UH5T2 and UH8T2, the difference is likely due to the fact that all their somatic mutations fall in diploid regions (Supplementary Table 3). While the CCF method only uses those mutations, we also account for CNV regions that, in these tumours, encompass a significant part of the genome (13% and 54%, respectively, Supplementary Table 5).

Sample S6T2 has no CNV regions (Supplementary Table 5). Therefore, the minimal discrepancy between the two methods is not due to the fact that we remove mutations in CNV regions.

We added this further analysis in the revised text (p.8 and Supplementary Fig.9).

4. It is also worth noting that the cancer genes classified as "drivers" in the above analyses may function differently depending on the context in which they occur. To avoid confusion, these should be referred to as cancer genes.

Response: We agree with the reviewer and modified this throughout the manuscript accordingly.

5. Returning to Table S1, why is it that some MSI-high tumors, which are typically diploid have such high numbers of copy number altered regions? For example Pt S6 (Lynch) has 3546 CNV genes in T1 and only 2 in T2. This is curious.

Response: Here the reference to Table S1 is incorrect and the reviewer likely refers to Table 1. It should be noted that what we report in Table 1 are not copy number regions, but copy number genes. We are interested in CNV genes and not in CNV regions because our focus is the comparison of altered genes between paired lesions.

Overall, we observed that hypermutated CRCs (including MSI-h) have less amplified and deleted genes than non-hypermutated tumours (Fig. 1a). This is in agreement with the literature and with the expectation of the reviewer. From Fig.1a it is clear that also S6T1 has a much lower fraction of CNV genes than non-hypermutated samples. Moreover, we cannot find any evidence suggesting an overestimation of the CNV genes in S6T1 or an underestimation in S6T2.

6. The classification of alterations as clonal, biclonal, polyclonal is rather non-conventional although there may be value in this simplified view. Nonetheless, it should be stated that neither subclonal copy number or clonal genotypes can be resolved from these data and that the density distributions are intended to provide a crude summarization of the expected number of alterations of a given class.

Response: The fact that the density distributions provide an estimate of the expected number of alterations at each clonality is stated in the text (p.7). We now further clarified this in the Supplementary Note.

7. The prediction of mutation pathogenicity is a challenging task and it is generally accepted that no single method has optimal performance across an array of functional categories or architectures. The authors employ a number of well established tools for this prediction, and take a union set approach by calling variants as damaging if they are categorized as such by 4/5 methods. This is a reasonable strategy, but it may be worth considering whether more recent methods such as CADD [PMID: 24487276] or DANN (a deep neural network version of CADD) [PMID: 25338716] both of which integrate diverse annotations identify additional high-confidence variants.

Response: The damaging effect of mutations has been used for the analysis of germline SNPs in MEGA (p.9 and 32-33), which constitutes the most innovative part of our work. Therefore, we wanted to be very conservative and retained only high confidence damaging mutations supported by multiple methods (at least four, p. 32). Indeed, it is crucial that the predictions are well supported since we use this information to identify biological processes that are constitutional altered in syCRC patients.

Before considering new damaging mutations from the methods suggested by the reviewers, we checked whether they were in fact high confidence variants.

To this aim we predicted the damaging effect of rare germline missense mutations using CADD (as a representative of meta-methods) in the 10 CRC patients sequenced in this study. The results are reported below.

Sample	Missense germline mutations (MAF < 1%)	Prediction of damaging germline mutations						CADD damaging mutations (phred>15)		
		SIFT	Poly Phen 2	Mutation Tester	Mutation Assessor	LTR	Supporte by at least 4 methods*	All	Already in our set	New and supported by at least 4 methods
S13N	300	96	129	183	8	105	41	123	38	2
S6N	283	92	100	150	6	89	26	103	23	1
S3N	303	106	121	170	13	99	39	124	34	6
S12N	272	85	105	152	5	87	33	107	29	2
UH1N	259	77	92	156	5	80	27	100	26	1
UH2N	391	113	122	223	13	124	34	142	30	2
UH5N	290	78	109	161	8	86	30	110	26	2
UH6N	247	77	85	162	6	78	24	101	24	4
UH8N	356	104	126	201	8	107	44	127	41	0
UH11N	475	161	156	220	11	120	45	134	43	2

* = final set of damaging mutations considered as an input for MEGA (Supplementary Table 7).

Based on these results, we can make a number of observations:

1- the initial number of damaging mutations predicted by CADD in each sample is similar to SIFT, PolyPhen etc. This suggests that there may also be poorly supported predictions in CADD and we cannot simply add the new variants, but need to verify whether they have the support of other methods;

2- the vast majority (~92%) of mutations in our final set are also predicted as damaging by CADD. This confirms that we retain highly supported predictions;

3- only very few new predictions made by CADD and not present in our final set are supported by other methods. Therefore, we do not miss additional high confidence mutations.

The comparison with other methods shows no evidence that the new methods will add new high confidence damaging variants and confirms that the combination of several approaches is most powerful to avoid poorly supported predictions.

Reviewer #4 (Remarks to the Author):

*1. Many of the statistical methods seemed ad hoc with assumptions about direction without a good rationale (e.g., **one-sided Wilcoxon test**) or convenient (**stratifying gene expression rather than applying differential expression analysis**). An FDR of 0.1 is reasonable, but also generous when a few gene sets are identified and the authors **don't report the statistics for enrichments (e.g., OR and p-values)**. How do the authors deal with the **uncertainty in estimating cell counts from slides**?*

Response: We respectfully disagree with the reviewer that the methods were chosen *ad hoc* but we acknowledge that we did not explain the rationale to justify our choice. In particular:

One-tailed Wilcoxon test: We applied one-tailed test in MEGA because we specifically searched for biological processes or disease-associated gene sets that were enriched in inherited damaging mutations. We now explicitly explain this in the text (p.9) and in the Methods (p.33).

To verify that we did not overestimate the significance of enrichment by using one-tailed test, we measured the p-values using two-tailed tests and added the results in Supplementary Table 8. The biological processes and disease-associated gene sets previously identified as enriched were all confirmed as statistically significant (Supplementary Table 8).

FDR of 0.1. Although we set the threshold of significance to $FDR < 10\%$, the support of significance is much stronger in most comparisons (for example, all enriched gene sets are below 5% FDR, Fig. 3b). To acknowledge this, in each specific comparison we now report the highest FDR value instead of the general threshold of 10%.

Statistics for enrichments (e.g., OR and p-values): Original p-values of Wilcoxon test prior to FDR corrections were already reported in Supplementary Table 8. We now added the odds ratio of the Fisher's tests in Supplementary Table 9a.

Stratifying gene expression rather than applying differential expression analysis: We compared biological processes in terms of fractions of not expressed, lowly, medium and highly expressed genes to be consistent with the approach used in the analysis of inherited mutations. We now explain this further (p.12).

To verify that this does not lead to spurious results, we performed a more conventional gene set enrichment analysis of differentially expressed genes (Methods p.36). We found that all gene sets that were enriched in not expressed or lowly expressed genes in the previous analysis were also found as significantly down regulated in the second approach (Supplementary Table 9b). Moreover, additional immune-related pathways were also significantly down regulated (Supplementary Table 9b). This further analysis strengthens our results and we added it to the

manuscript (p.12-13, 36, Supplementary Table 9b).

Uncertainty in estimating cell counts from slides: The counts of immune cells from tissue slides was led by an experienced mucosa immunologist, Professor Jo Spencer, and by a colorectal cancer pathologist, Dr Manuel Rodriguez, both of whom have several years of experience in the field. Moreover, they used well-established methods that they have previously published (refs. 28-29).

Each slide was assessed blindly to the tumour details (syCRC, soCRC, MSI and MSS) or to the origin of the normal colonic mucosa from syCRC or soCRC patients.

Although we acknowledge a certain degree of uncertainty that is proper in this type of analysis, we think that we accounted for all possible factors to minimize it.

*2. The authors set out to conduct a systematic exploration, yet restrict the gene sets to kegg only, and examine only a couple immune cell types. Did the authors consider **neutrophils and lymphocytes separately and not find differences**? Also, the term **constitutional alteration** seems a bit strange and maybe too strong given the data. Also, I don't understand how the authors claim **deregulation from the transcriptional data**. The transcriptional levels are different, but it's not clear whether this is deregulation or some other mechanism.*

Response: The reviewer here touches upon different issues namely:

Restriction of gene sets to KEGG only: We do not restrict the gene set analysis to KEGG pathways only, but apply MEGA to disease-associated genes from GWAS as a further confirmation (p.10, Fig.3 b-d; Supplementary Table 8).

Analysis of neutrophils and lymphocytes separately: We show the neutrophil-to-lymphocyte ratio because this parameter has a well-known systemic prognostic value in colorectal cancer¹⁶⁻¹⁸.

When neutrophils and lymphocytes are analysed separately, we observe a significantly higher number of neutrophils in syCRCs while the number of lymphocytes is comparable (Supplementary Figure 12). We comment on this in the text (p.12).

Constitutional alteration seems a bit strange and maybe too strong given the data: The definition 'constitutional alteration' was used not to comment on our results but to introduce the rationale of MEGA on germline mutations (p.9).

Deregulation from the transcriptional data: We agree with the reviewer and modified the text accordingly (Abstract and p.14).

*3. One hypothesis that wasn't explored, but would be interesting is what sort of environmental perturbation might trigger the cancer. For example, the authors could examine **viral integration or the propensity of viral RNA in their samples to determine whether the germline mutations in immune-related genes predispose an individual to develop an oncogenic conversion following some sort of pathogenic or chemical perturbation that exploits the immune-related pathways (i.e., Toll)**. The presence of the immune cells are somewhat speculative and one cannot determine whether they are drivers or passengers. (my guess is the latter)*

Response: This is a fascinating hypothesis that is unfortunately hard to test with available data. The analysis of viral integration would require whole genome sequencing data, which we do not

have. In addition, there are no clinical records of ongoing viral infection at the time of resection in these patients.

We speculate that they modify the immune response already in the normal mucosa and we find independent evidence that this may be the case because the immune cell composition is different.

4. The paper is well organized and the arguments are clear. The authors need to fix some typographical and grammatical errors. For example, the abstract could be improved through some clear language and edits. I've made one change below.

"This inter- and intra-tumour heterogeneity has consequences for identifying effective treatments and monitoring resistance. To understand the causes of syCRCs, we searched for biological processes that are altered in syCRC patients compared to patients with solitary colorectal cancer or to healthy individuals."

Response: We have modified and fixed the Abstract along the lines proposed by the reviewer.

References

1. Carter SL, *et al.* Absolute quantification of somatic DNA alterations in human cancer. *Nat Biotechnol* **30**, 413-421 (2012).
2. Van Loo P, *et al.* Allele-specific copy number analysis of tumors. *Proc Natl Acad Sci U S A* **107**, 16910-16915 (2010).
3. George J, *et al.* Comprehensive genomic profiles of small cell lung cancer. *Nature* **524**, 47-53 (2015).
4. Brastianos PK, *et al.* Genomic sequencing of meningiomas identifies oncogenic SMO and AKT1 mutations. *Nat Genet* **45**, 285-289 (2013).
5. Jones S, *et al.* Personalized genomic analyses for cancer mutation discovery and interpretation. *Sci Transl Med* **7**, 283ra253 (2015).
6. Schiavon G, *et al.* Analysis of ESR1 mutation in circulating tumor DNA demonstrates evolution during therapy for metastatic breast cancer. *Sci Transl Med* **7**, 313ra182 (2015).
7. Stachler MD, *et al.* Paired exome analysis of Barrett's esophagus and adenocarcinoma. *Nat Genet* **47**, 1047-1055 (2015).
8. Brastianos PK, *et al.* Exome sequencing identifies BRAF mutations in papillary craniopharyngiomas. *Nat Genet* **46**, 161-165 (2014).
9. McFadden DG, *et al.* Genetic and clonal dissection of murine small cell lung carcinoma progression by genome sequencing. *Cell* **156**, 1298-1311 (2014).
10. Crompton BD, *et al.* The genomic landscape of pediatric Ewing sarcoma. *Cancer Discov* **4**, 1326-1341 (2014).
11. Guinney J, *et al.* The consensus molecular subtypes of colorectal cancer. *Nature medicine* **21**, 1350-1356 (2015).
12. Baca SC, *et al.* Punctuated evolution of prostate cancer genomes. *Cell* **153**, 666-677 (2013).
13. Eleveld TF, *et al.* Relapsed neuroblastomas show frequent RAS-MAPK pathway mutations. *Nat Genet* **47**, 864-871 (2015).
14. Paik PK, *et al.* Next-Generation Sequencing of Stage IV Squamous Cell Lung Cancers Reveals an Association of PI3K Aberrations and Evidence of Clonal Heterogeneity in Patients with Brain Metastases. *Cancer Discov* **5**, 610-621 (2015).

15. McGranahan N, Favero F, de Bruin EC, Birkbak NJ, Szallasi Z, Swanton C. Clonal status of actionable driver events and the timing of mutational processes in cancer evolution. *Sci Transl Med* **7**, 283ra254 (2015).
16. Pine JK, *et al.* Systemic neutrophil-to-lymphocyte ratio in colorectal cancer: the relationship to patient survival, tumour biology and local lymphocytic response to tumour. *Br J Cancer* **113**, 204-211 (2015).
17. Neal CP, *et al.* Prognostic performance of inflammation-based prognostic indices in patients with resectable colorectal liver metastases. *Med Oncol* **32**, 144 (2015).
18. Malietzis G, Giacometti M, Kennedy RH, Athanasiou T, Aziz O, Jenkins JT. The emerging role of neutrophil to lymphocyte ratio in determining colorectal cancer treatment outcomes: a systematic review and meta-analysis. *Ann Surg Oncol* **21**, 3938-3946 (2014).

"

"

REVIEWERS' COMMENTS:

Reviewer #2 (Remarks to the Author):

The author's have now addressed my concerns.

Reviewer #4 (Remarks to the Author):

The authors did a nice job in their point by point response and the updated manuscript to address the reviewer's concerns.